# Immersive Virtual-Reality System for Aircraft Maintenance Education: A Case Study

Águeda Gómez-Cambronero, Ignacio Miralles , Anna Tonda and Inmaculada Remolar *

Institute of New Imaging Technologies, Universitat Jaume I, 12006 Castellón, Spain; cambrone@uji.es (Á.G.-C.); mirallei@uji.es (I.M.)
* Correspondence: remolar@uji.es

**Abstract:** Aircraft maintenance is a highly relevant procedure in many industries, yet obtaining qualified personnel to carry it out is a difficult task. Training in such techniques is complex and requires access to facilities and materials that are not readily available. Virtual reality can be a tool to improve this situation. This paper presents the whole process of design, development, and evaluation of a virtual environment that allows users to perform some of the main tasks required in aircraft maintenance after landing or for take-off. By following a user-centered design methodology and the Octalysis framework to apply motivation and engagement techniques, a gamified virtual environment was developed that allows the user to practice specific aircraft maintenance techniques. The environment was tested by users of different profiles who answered questionnaires to evaluate the perceived gamification, usability, and the feeling of sickness from the experience. The analysis of the data corroborates the good performance of the VR environment in these fields.

**Keywords:** virtual reality; immersive learning; learning by doing; aircraft maintenance

## 1. Introduction

Aircraft maintenance is an essential procedure to preserve the reliability and performance of the aircraft. Despite the fact that this maintenance represents approximately 9–15% of airlines' operating costs [1,2], the entire aviation industry is still facing a shortage of specialized personnel who can perform mechanical tasks, as noted by Flying magazine [3]. Carrying out this training requires access to costly equipment, as well as the support of an experimental trainer, which may not be affordable for all training centers, so learning-by-doing methodology plays a vital role in this field. Currently, the training procedure is conducted within the confines of the hangar whenever an operational aircraft requires maintenance. The traditional approach involves combining theoretical classes using dossiers with practical exercises where students can apply what they have learned in a real-world setting. However, the practical exercises are subject to very restrictive and limited conditions, making the learning process both arduous and expensive. Virtual reality (VR) allows for experiencing close-to-reality situations in safe and immersive 3D environments. VR has been shown to be an effective tool for enhancing technical skills and practical learning [4–7]. Due to its inherent characteristics, this technology is well-suited to support a wide range of learning methodologies, including experiential learning. Moreover, VR can enhance student engagement and motivation, offer personalized learning experiences, and facilitate collaborative learning through its capability for simultaneous access.

In comparison to traditional methods, the use of VR technology for training presents both advantages and disadvantages. To utilize VR technology effectively, one must first model the environment, tools, and components required for practice, as well as programming specific tasks. This modeling is dependent on the particular aircraft model and location, which may present challenges with scalability and generalization of practices. In addition, specialized hardware and knowledge of the technology are necessary. Moreover, prolonged use of VR simulation may cause adverse side effects, making long-lasting

practices inadvisable. However, there are several noteworthy advantages to using VR for training. Firstly, students are no longer limited by time or location as VR enables them to practice in a simulated environment that mimics real-world conditions. Secondly, VR technology provides safe and personalized training tailored to each student's individual requirements, allowing for endless repetition of exercises without negative consequences. Thus, the benefits of regular practice are preserved while providing an improved learning experience.

Introducing gamification in VR experiences can enhance the effectiveness of learning by creating engaging environments that motivate students, foster collaboration, and provide immediate feedback. Gamification involves taking the enjoyable and engaging aspects that make games fun and applying them to non-game contexts [8]. One of the common mistakes when gamifying a system is to focus it on the pontification strategy due to the misconception that it is solely linked to the use of points–badges–leaderboards (PBL) [9]. However, gamification is a design methodology focused on the users, their motivations, and their behavior. Octalysis [10] is a gamification framework based on the drives of the user's motivation. It uses an octagon shape to propose an analytical framework that associates each of its sides with a Core Drive, such as Epic Meaning, Accomplishment, Empowerment, Ownership, Social Influence, Scarcity, Unpredictability, and Avoidance. This framework also distinguishes between Core Drives that are considered extrinsic or intrinsic motivators, as well as positive or negative motivators, through their positioning in the octagon shape. Specifically, Core Drives located on the left of the octagon are classified as extrinsic motivators due to their relation with logic, calculations, and ownership, while those on the right are considered intrinsic motivators for being associated with creativity, self-expression, and social aspects. Additionally, Core Drives at the top of the octagon are viewed as positive motivators, while those at the bottom are seen as negative motivators [10].

This article presents the design process and evaluation of a playful learning experience designed to perform aircraft maintenance training that has been gamified using the Octalysis framework. The experience was designed to take into account the needs and opinions of the actors involved in this learning process. The assessment serves as an initial gauge of the feasibility of the user experience and acceptance, predicated on the perceived gamefulness, ease of system use, and absence of negative health outcomes. This gameful learning experience offers an immersive and realistic 3D environment in which students can acquire specific aircraft maintenance knowledge through a close reality practice in a safe mode. The objective of this application is to take advantage of the benefits of VR to offer a cost-effective and successful way for aircraft maintenance training.

There are currently commercial solutions in the field of aircraft maintenance that use VR and augmented reality (AR) technology. Boeing has developed an AR app that is used with smart glasses to facilitate complex maintenance procedures for electrical technicians. By overlaying digital instructions and illustrations onto the technician's field of view, the app facilitates the identification and repair of components. Another solution is Airbus HoloLens App, which provides maintenance operators and cabin crews with access to information and instructions while they are on the job. While these solutions facilitate maintenance tasks, the first one is not designed for training purposes. Additionally, since they are AR applications, they can only be used in the real context, which limits their use. Lufthansa has also implemented VR training, which replicates a real-flight environment and trains flight attendants. Finally, Honeywell has developed VR simulations aimed at optimizing the training of plant operators and field technicians. These simulations include a digital replica of the physical plant, enabling workers to receive skill-based training. The personalized virtual environment allows workers to familiarize themselves with the plant and its operations. While the latter two solutions are similar to the approach presented in this article in that they utilize a digital replica of a realistic training environment, the first solution is specifically focused on cabin crew training. However, none of them provided specific indications on maintenance tasks or solutions to encourage the autonomous utilization of the platform.

The rest of the work is organized as follows: Section 2 briefly reviews important VR applications on aircraft maintenance and education. Section 3 describes the methodology followed, including the design process, implementation, and design of the study. Section 4 exposes the hypotheses set out in relation to the experience created and the tests that were carried out. Next, the obtained results are analyzed and discussed in Section 5, and finally, the conclusions and future work are presented in Section 6.

## 2. Literature Review

Learning by doing [11] is a methodology that has been demonstrated as efficient in the learning process. It is based on learning from experiences that result directly from one's own actions. In other words, it is a method by which learners obtain the most out of their education through active participation. The goal is to give future professionals a chance to apply the knowledge and skills they have learned in a real-world setting, which can help them become more proficient and confident in their abilities. Additionally, learning by doing can help reduce the time and cost associated with traditional training methods.

Regarding professional training, the usual problem for learners is that accessing the physical material or environments where obtaining knowledge is generally very difficult and expensive. In these cases, the digital representation of the training environments can be the solution. Some papers related to this topic support this assertion, such as the one presented by Grandi et al. [12]. In this work, they presented an application of VR technologies to create virtual training simulations addressing assembly or maintenance tasks. The paper suggests a methodology to create an interactive virtual space in which operators can perform predefined tasks in a realistic way, with dedicated instructions to support the learn-by-doing based on key training features. The results show that operators generally appreciate this new training process, enabling faster and more intuitive learning.

Aircraft maintenance is one of the fields that have more difficulties because the learners have to demonstrate a high level of expertise to access the planes. Therefore, beginners are not generally allowed to access the hangars to practice their knowledge, although they have to obtain competencies to ensure error-free maintenance. Vora et al. presented in [13] a work that measured the degree of immersion and presence felt by subjects in a virtual environment simulator. Specifically, it conducted two controlled studies using the VR system developed for the visual inspection task of an aft-cargo bay. The results of this study indicated that the system scored high on the issues related to the degree of presence felt by the subjects. In 2008, Bowling et al. presented [14] work on training aircraft maintenance inspectors. The purpose of their study was to develop a simulated aircraft cargo bay in a virtual reality (VR) environment to explore VR as a training tool and examine differences between general and detailed inspection under paced and unpaced conditions. Eschen et al. presented in [15] a concept to evaluate the potential of inspection and maintenance processes in the aviation industry regarding the use of mixed reality systems. Four different use cases were discussed applying augmented or virtual reality devices in an industrial context.

More recent works in this field were presented by Lee et al. [16]. They proposed a system that incorporates virtual reality methods into the classroom to compensate for the shortcomings of the existing remote models of practical education. Based on the proposed system, the authors developed an aircraft maintenance simulation and conducted an experiment comparing their system to a video training method. Some retention tests were conducted, and presence was investigated via survey responses. The ones given to the presence questionnaire confirmed a sense of spatial presence felt by the participants, and the usability of the proposed system was judged to be appropriate. In the same year, Wu et al. [17] presented in their work an aircraft maintenance virtual reality system. For a Dornier-228 aircraft, a walk-around visual inspection of its fuel system was designed in a virtual environment. The environment was tested by students to validate the effectiveness of using the system in training. The students acknowledged that the system was beneficial

to their learning, which proved that the developed system is highly effective for training 148 students to improve aircraft maintenance skills.

Despite the fact that the innovative characteristics of these types of systems make them very attractive, it is relevant to consider the aspects of motivation and adherence so that their acceptance is adequate. Gamification has been widely used in VR simulations to make them more attractive and encourage their adherence by incorporating elements such as points, badges, leaderboards, and rewards. Lampropoulos et al. [18] stated in their work that using augmented reality and gamification in education can yield several benefits for students and assist educators. In work presented by Falah et al. [19], gamification was applied to train chemistry students in complex content, demonstrating its acceptance and the desire of students to integrate it into regular classes. Similar results were obtained in [20], providing teachers and students with techniques to learn how to use search algorithms. Villagrasa et al. [21] applied these techniques in the training of architecture students, again with results that show that motivation and adherence improve with them.

From a perspective less focused on the student and more on the professional world, Ulmer et al. [22] compared a gamified virtual environment with the real one. The environment consisted of a manual work station with assembly parts, an Allen key, and a monitor showing a picture of the final assembly. The training presented to two groups was compared, and it was shown that VR reduced error rates and the need for support. The same results were obtained by Suncksen et al. [23], who presented a simulation-based training system to support the training in x-ray imaging of operating room personnel in a virtual environment. As a way of structuring the use of gamification, Octalysis [10], the framework mentioned above, has been used in different works. In [24], Ewais et al. used the framework to classify 12 stress management mHealth apps. Marisa et al. [25] relied on it and the k-means clustering approach to explore which drives most influence customer behavior for the Customer Lifetime Value indicator. In learning, Irawan et al. [26] used Octalysis to design and build a React Native learning app and measure the level of intention to use the behavior.

By taking all these reviews into account, this work presents a method based on the learning-by-doing methodology addressed to the representation of gamified VR environments related to aircraft maintenance. It represents a virtual environment that motivates students to take action in the VR scenarios to investigate and test the different actions to be performed in this field. All the requirements and the user design environment are analyzed in the following sections.

## 3. Materials and Methods

The process of design, implementation, and validation was executed by a multidisciplinary team following an iterative process. The tasks that were performed are described subsequently.

### 3.1. Design Process

The design process was carried out following a user center design (UCD). This process seeks a design focused on gaining a deep understanding of end-users. In order to meet their needs and validate the good performance of the designed virtual environment, the authors collaborated closely with Castelló Airport (Aerocas), the company that manages its maintenance, and the training of students in the field (Brok-Air). Through meetings with the parties involved, the training needs and the issues faced by this teaching were gathered. Design thinking (DT) was used to guide the UCD. DT is an approach to problem-solving that involves empathizing with users, defining problems, ideating solutions, prototyping, and testing. Specifically, in the development of this experience, it took into account the five DT principles in the following manner:

1.  **Empathize:** the user's perspective (students, teachers, and professionals) was attended in different meetings with the observation of the professional aircraft maintenance work.

2.  **Define:** specific and clear requirements were determined, ensuring the alignment of the development process with the real needs of end-users.

3.  **Ideate:** multiple ideas were generated to address the specific challenge. Through brainstorming techniques, the team members from different fields were encouraged to think creatively and share their ideas.

4.  **Prototyping:** incremental prototyping was used, including new models and functionalities each time depending on the stage of the design process.

5.  **Testing:** assessment of the functionality was carried out in each development interaction. Furthermore, a study to measure usability, gameful perception, and sickness feelings of the application was conducted.

The primary challenge encountered during the training process was the issue of accessibility. Consequently, it became necessary to create a digital replica of the training environment. To facilitate iterative prototype development, specific learning tasks that were currently being taught in the program were selected. These tasks included the installation of landing and removing gear downlock pins, as well as the manipulation of height and temperature sensors. Following successful implementation and approval by the trainers, subsequent tasks in the course were then incorporated. Additionally, the solution had to promote the habit of practice among trainees. Therefore, all identified requirements were properly integrated into the system.

After the elaboration of the requirements, it was discovered that the targeted knowledge could be acquired through the repetitive execution of the specified tasks. In order to minimize boredom and promote engagement in the learning process, the experience was gamified through the Octalysis framework [10]. The experience was gamified by attending to each Core Drive in the Octalysis framework as follows:

*   **Meaning**: The meaning allows the user to feel that the actions to perform have a general objective. In the case of this gameful experience, the objective is to prepare a plane after its landing process (carrying out two specific actions) or prepare it to take off (using two other actions). In order to offer the user a more general meaning of this objective, a general missions screen (Figure 1) is presented where the user can see that their efforts are reaching a more general final result, taking advantage of this narrative to improve his motivation.

*   **Development and Accomplishment**: Aligned with the narrative that gives meaning to the user's actions is the achievement of partial objectives. In the gameful experience, these partial objectives are shown as a progress bar that allows the user to visualize the achievements (Figure 2). This progress bar indicates which actions have been successfully performed and the ones that remain undone to complete the mission. Moreover, the user is offered the possibility to repeat tasks in order to try to execute them more efficiently, reducing the time to solve the missions. This fact not only motivates the user to progress until all the objectives are successfully achieved but also encourages repeatability of the missions, improving their professional skills.

*   **Empowerment of Creativity and Feedback**: This Core Drive refers to the users' impulse to find different combinations of solving the proposed target and the feedback received lately. Although the problem to be performed has been previously established, the way it is solved can vary according to the user's own experience. It is given the possibility of moving freely around the scene and analyzing the different 3D models generated before performing the task. In order to promote engagement, the possibility of sharing the achievement on social networks is also offered (Figure 3). This provides an opportunity to receive feedback from the community suggesting new ways to explore the environment.

*   **Ownership and Possession**: The ownership feature is the one that is identified with the possession of goods or benefits of the user in the application. Some prizes were designed to accomplish this driver. They are a set of collectible miniatures of airplanes that are obtained by the user once per every task successfully performed (Figure 4). As the user overcomes different challenges, such as successfully completing the initial

tutorial, a 3D model of a specific plane is unblocked, losing its initial red color, and it is offered on the table that contains the collection. Aircrafts representing missed or failed tasks are colored in red to motivate the user to repeat missions or perform new activities in order to obtain the complete collection.

- **Social Influence and Relatedness**: Feeling accepted by a group is key to the motivation to carry out any activity, and that is characteristic of this axis. In the gameful experience, the user's interaction with the community is encouraged, allowing them to share the different achievements or unlocks that they carry out on social networks (Figure 3). This makes it possible to check the achievements of other users, comparing, commenting on, and valuing the different interactions carried out by them.

- **Scarcity and Impatience**: This type of feature is traditionally used in games where scarcity (of resources or time) conditions the game. In the educational context, depriving the user of access to learning content is not considered an effective tool. However, there are two features of the virtual environment that cover this axis of the methodology. On the one hand, there is the "Collection" section (Figure 4), where the user will see the planes they have not yet achieved colored in red. On the other hand, to access the new missions, it is mandatory to have passed the first ones, which generates a delay and reduced availability of content until the user achieves the previous objectives. The use of these tools encourages users to advance in the tasks, thus enhancing their skills.

- **Unpredictability and Curiosity**: Encouraging curiosity is a tool used in many games as a measure of attraction (and, in many cases, monetization): reward boxes or envelopes are some of the most commonly used examples. In this gameful experience, the possibility of collecting miniature aircraft is complemented by this feature. Each time the user successfully completes a mission, the aircraft provided to the user from the ones blocked is randomized (Figure 5), so if the user wants to obtain a different one, they could repeat the experience to try to obtain a different one. This technique is intended to encourage the re-playability of the scenarios, thus receiving more practice with the training content.

- **Loss and Avoidance**: Although the loss of content is a tool used in motivation, it was not considered to be positive in the training context. However, covering this drive, the gameful experience includes a "full" mode (Figure 6) where the user must pass all the phases in a row without abandoning or failing any of the missions. The development of the tasks complements the mode in which progress is stored, never going backward, and offers as a reward a miniature that cannot be obtained in any other way, thus providing extra motivation for users to repeat the scenarios and practice the training contents more times.

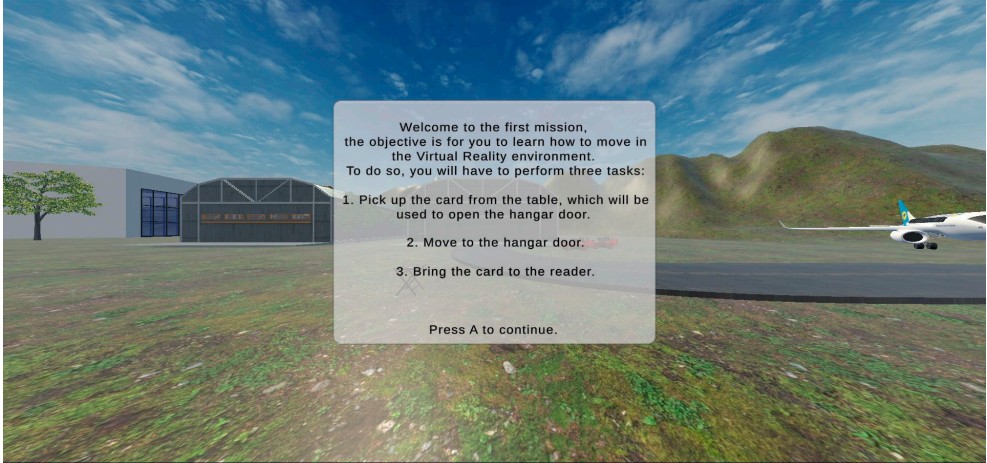

**Figure 1.** Narrative included in different scenes to enhance the meaning of the experience.

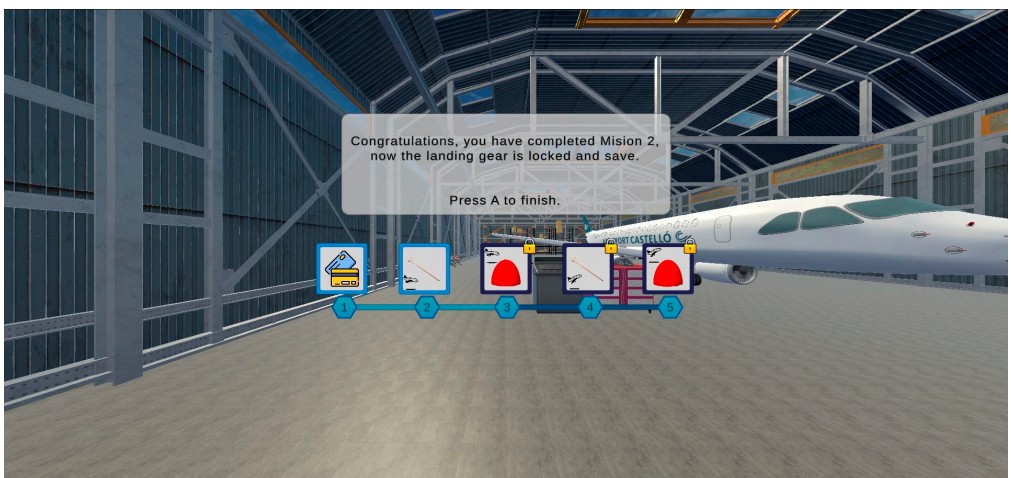

**Figure 2.** Progress bar indicating the completed missions and the ones that remain unsolved.

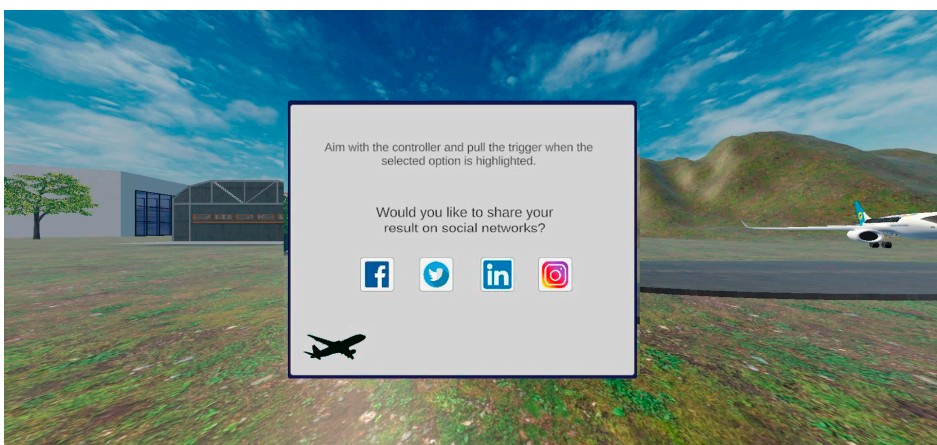

**Figure 3.** Connection with social networks to share user achievements.

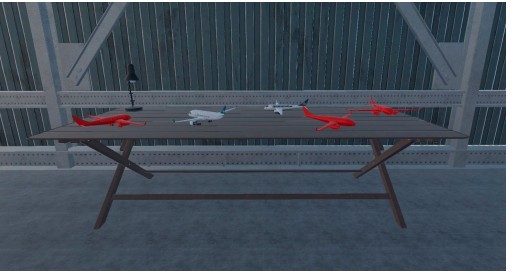 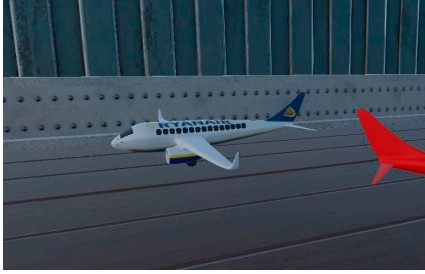

**Figure 4.** Mini-aircrafts offered as prize when some missions are successfully performed.

A graphical representation of the applied Octalysis framework and a summary of the game mechanics designed according to each Core Drive can be found in Figure 7. This figure was created using the web-based tool provided by Yu-kai Chou, the author of the Octalysis framework, which is available at https://www.yukaichou.com/octalysis-tool/ (accessed on 17 April 2023). As can be noted in this figure, the strength of this gamified system lies in extrinsic and positive motivators.

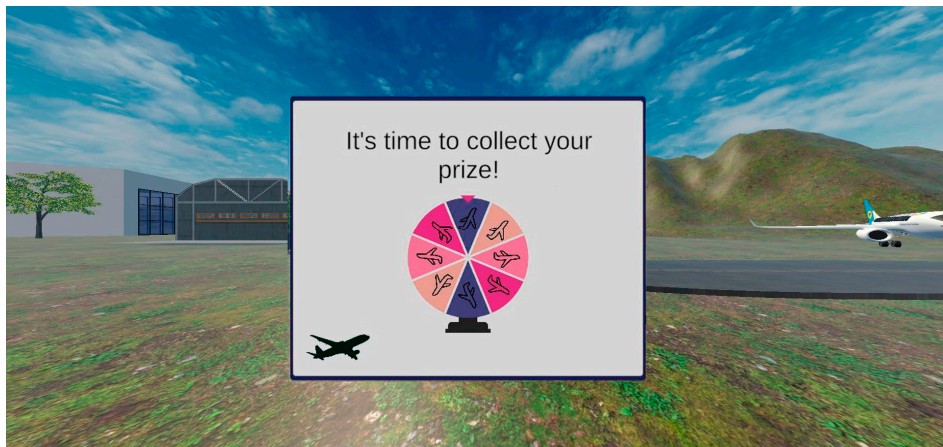

**Figure 5.** The lucky wheel represents the possibility of obtaining a miniature aircraft that could be a new or a repeat one.

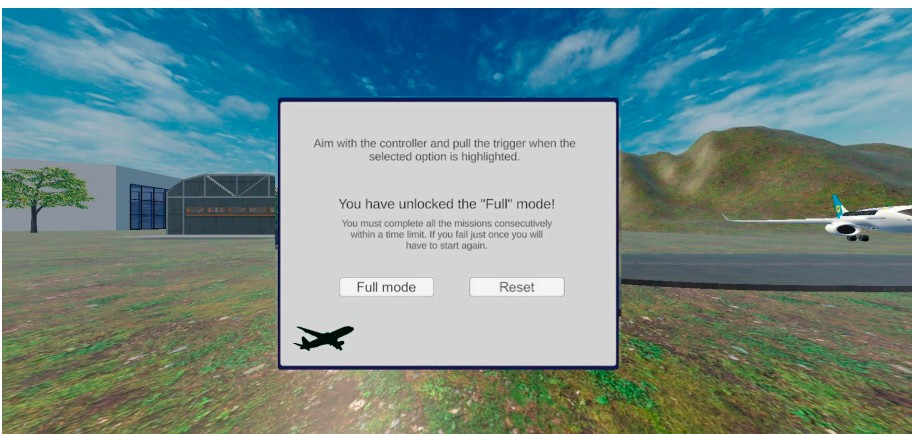

**Figure 6.** Full mode allows the user to face the challenge of completing all the missions in a row without fail.

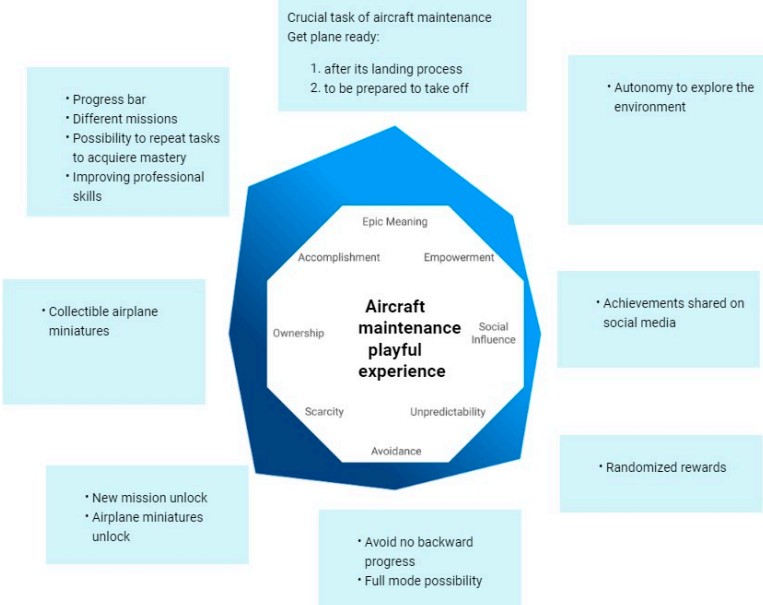

**Figure 7.** Applied Octalysis framework.

*3.2. Gameful Experience*

Joining the analysis of the main requirements of the gameful experience extracted after the user-centered design process and the application of the Octalysis framework, the details of the implemented experience were established. They are described next.

### 3.2.1. Technical Aspects

In order to develop the complete gameful experience, Unity 3D video game engine (version 2020.3.29f1) was used, taking advantage of the Open XR virtual reality libraries (including the Interaction XR Toolkit) to connect the development with the output device: Oculus Quest 2. The decision was made to use Unity 3D for the development of the experience due to its lower system requirements in comparison to Unreal Engine, which was also considered as an alternative. The scenes and 3D elements were modeled using Blender. All development was performed with an Inter(R) Core(TM) I7-10700F CPU, including an Nvidia GeForce RTX 3070 Ti graphics card with 8GB of memory.

As shown in Figure 8, the system runs on a high-performance computer that supports the game engine. Once launched, the environment presents the models, with the gamification and gameplay designed, allowing the user to interact through VR devices. This interaction makes the system present the different challenges of the experience, providing them with greater knowledge about the techniques described.

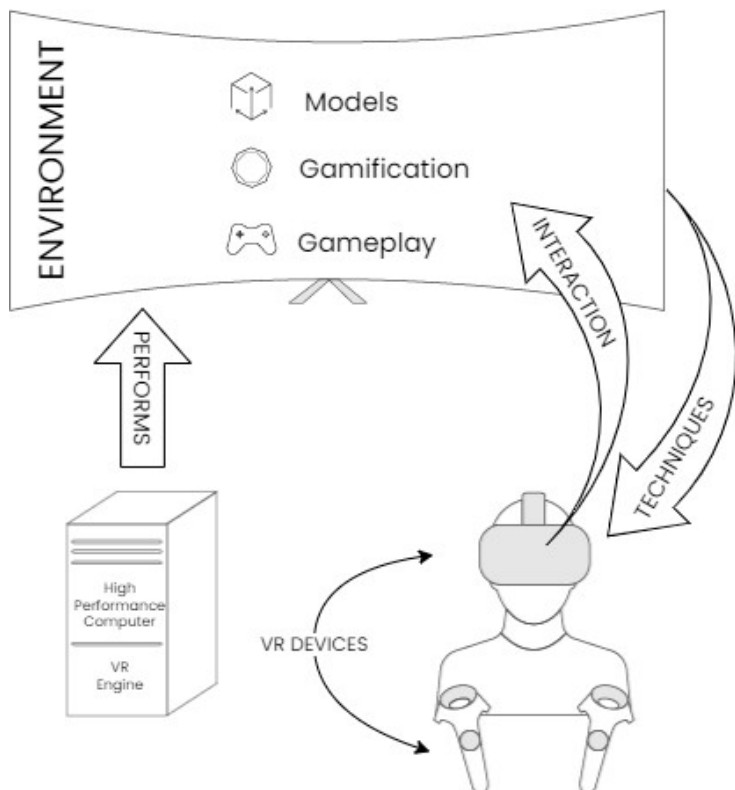

**Figure 8.** Gameful experience.

### 3.2.2. VR Experience

The tasks necessary to carry out the maintenance of the aircraft were determined by the staff of Castelló Airport (Aerocas). With the purpose of providing students with the opportunity to practice with experiences close to reality, the entire real-world environment was replicated. Figure 9 shows samples of the result.

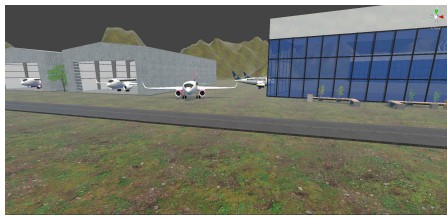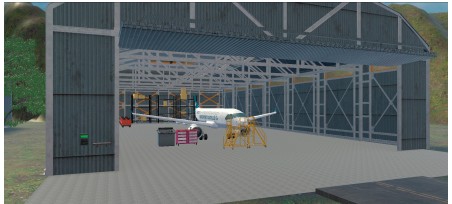

**Figure 9.** General views of the scenario representing the Castelló Airport (Aerocas).

Four operations, which are necessary for students to practice during their training with the Boeing 737 Next Generation, were simulated as follows:

- Landing gear downlock pins installation.
- Hiding height and temperature sensors.
- Removing gear downlock pins installation.
- Unhiding height and temperature sensors

The simulation starts when the aircraft has landed, and its maintenance has to be performed at the hangar. From that moment, the user is faced with five challenges that must be overcome in a specific order, although they can repeat those that they have previously completed.

The **first** challenge is designed as a training scene, where an introduction to the virtual environment and its main interactions are presented. In order to become acquainted with the application management, the user has to move outside the hangar area, reach the specific hangar door, pick up a card, and open the door by using it. The **second** mission is already carried out inside a hangar where the user has to secure the landing gear by performing the landing gear downlock pins installation maneuver. The **third** mission consists of protecting the aircraft's temperature and altitude sensors with protective hoods. Therefore, users have to use a ladder that facilitates access to them. For the last two missions, the narrative changes; instead of receiving the landing of the plane, the user has to prepare it for take-off. In the **fourth** mission, the landing gear pin has to be removed, and, finally, in the **fifth** mission, the objective is to uncover the sensors.

Figure 10 shows details of the digital environment and objects that were modeled to perform these processes.

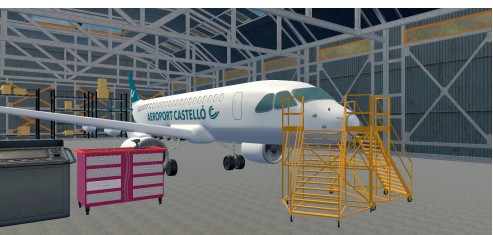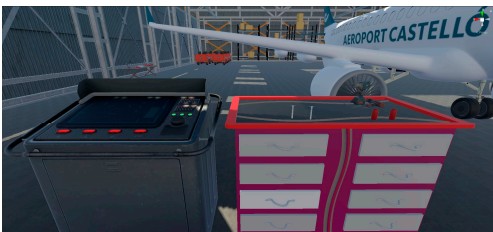

**Figure 10.** Details of the objects involved in the missions that were simulated.

## 4. Study Design and Scope

The scope of this work is to evaluate concepts related to gameful perception, usability, and sickness feelings. Three working hypotheses were defined:

**H1:** *The gamification of the gaming experience encourages the practice of the processes required in aircraft maintenance.*

**H2:** *The usability of the designed play experience facilitates the practice of aircraft maintenance tasks.*

**H3:** *The VR experience does not produce any discomfort for users during the experience.*

### 4.1. Participants and Procedure

The experiment involved 22 participants with an average age of 26 years (minimum age = 21 years; maximum age = 56 years; 50% women, 50% men). The presence of any disorder was considered an exclusion criterion for participating in the experiment. However, all participants in the study met the eligibility criteria and were not excluded on this basis. The individuals who participated in the study held technical or engineering undergraduate degrees, as well as master's or doctoral degrees. Furthermore, the participants confirmed their familiarity with video games and a high level of computer literacy.

At the beginning of the study, the participants were provided with information about the purpose and procedures of the research. They were also given a consent form to review and only provided their consent to participate in the study after they had understood the information and had their questions answered. Participation in the study was entirely voluntary, and refusal to participate had no negative repercussions. Despite all of them indicating that they had prior experience with video games, they were not regular players. Additionally, the participants confirmed that they did not have any preexisting disorders.

The study was conducted at the facilities of the Institute of New Image Technologies, extra-curricular. At the beginning of the experience, they received a brief introduction to the system and an explanation of the controls. Afterward, the participants started interacting with the application for approximately 15 min and then filled out quantitative questionnaires about sickness feelings, game experience, and the usefulness of the application. The mean duration for responding to the survey was 10 min. The questionnaires were filled out anonymously on the Qualtrics survey platform and stored on an encrypted server.

### 4.2. Evaluating the Experience

Different concepts were analyzed during the experience, measured by means of validated questionnaires presented in the literature. Three surveys were answered by every user after the VR experience in order to evaluate each proposed hypothesis. Each of them evaluates the feelings of the user regarding three different concepts: perceived gamefulness, usability, and sickness feelings.

#### 4.2.1. Perceived Gamefulness

The Gameful Experience Questionnaire (GAMEFULQUEST) [27] is an instrument designed to assess the degree of perceived gamefulness when using a system. GAMEFULQUEST consists of 30 items rated on a seven-item Likert scale (ranging from 1 "strongly agree" to 7 "strongly disagree"). The detail of this questionnaire is included in Appendix A. The highest score on this scale represents a better outcome. The items are divided into seven different dimensions: accomplishment, challenge, competition, guided, immersion, playfulness, and social experience. Both items and dimensions were randomized, as the authors note, in order to address the order and effect bias of the common method. The GAMEFULQUEST demonstrated good reliability (Cronbach's alpha of .87–92) [27]

#### 4.2.2. Usability

The System Usability Scale (SUS) [28] is a widely used tool for measuring the usability and overall user satisfaction of a technology. The SUS is a reliable questionnaire (Cronbach's alpha of .91) [29] that consists of ten items that assess different facets of system usability, including requirements for support, training, and complexity of use. Each of which is rated on a five-item Likert scale (ranging from 0—"strongly agree" to 4—"strongly disagree"). The detail of this questionnaire is included in Appendix C. In order to compute the scale score, the numerical responses for each item are added together, given that the items on the scale do not possess independent validity. The score for odd-numbered items is obtained by subtracting 1 from the response prior to summation, while for even-numbered items, their sum is subtracted from 5. The summation is then multiplied by 2.5, yielding a total score ranging from 0 to 100. Scores that fall below 68 are considered to be indicative of an unusable system.

### 4.2.3. Sickness Feelings

The Simulator Sickness Questionnaire (SSQ) [30] provides a quantitative measure of the severity of simulator sickness symptoms through 16 items, each rated on a 4-point scale ranging from 0 (none) to 3 (severe). The detail of this questionnaire is included in Appendix B. The 16 items belong to three types of symptoms: nausea, oculomotor, and disorientation. The minimum possible score is zero being the best-case scenario. SSQ indicates an adequate internal consistency (Cronbach's alpha of 0.7) [31].

## 5. Results and Discussion

The data obtained through the Qualtrics platform were managed in CSV format with Microsoft Excel File. The analysis was conducted by one of the authors and revised by the rest. No discrepancies appeared.

### 5.1. Perceived Gamefulness

The first hypothesis of the study states that the gamification of the environment motivated its use in training practices related to aircraft maintenance. To evaluate this information, the Gamefulness Questionnaire was used. Figure 11 shows the results obtained after the evaluation, including the mean and the standard deviation of the data.

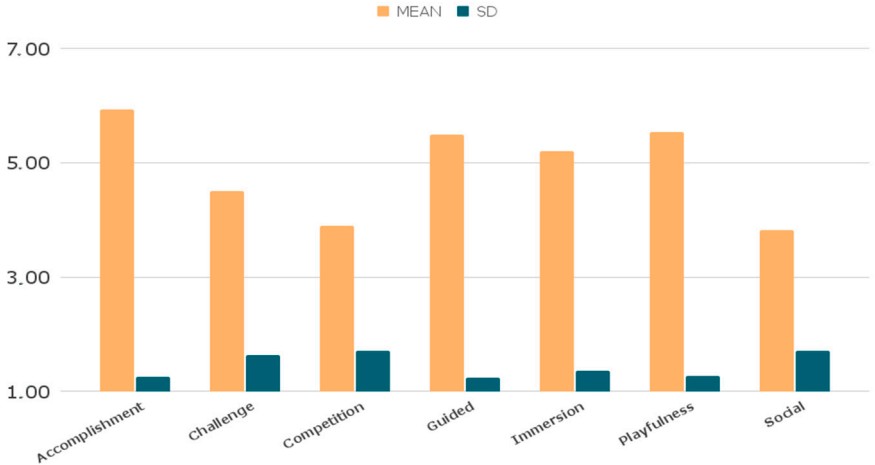

**Figure 11.** Gamefulness questionnaire results.

Analyzing the obtained results, the questions related to **accomplishment** are those with the highest score, reaching almost 6 (5.93). Fifty-seven percent of the participants have indicated that they strongly agree with "Makes me feel that I need to complete things", while forty-seven percent stated that they were clear about the objectives of the environment.

The environment was not excessively **challenging** for users, with an average score of 4.5. Only 24% agreed that the system pushed their limits; however, 53% strongly agreed or agreed that the system allowed them to work at a level close to what they were capable of. Similar to the challenge, the **competition** is one of the sections with a low score, a 3.9. Thirty-eight percent strongly disagreed that the system makes them feel like they are in a race. Moreover, 33% stated that they disagree or completely disagree that they want to be first. Users indicated that they feel **guided** using the system, with an average score of 5.59. Specifically, 86% of them agreed or completely agreed with the phrase "Makes me feel guided", and 62% agreed that the system gives them the feeling of having an instructor. **Immersion** is another aspect where the value is above 5 (5.21); 86% agreed or completely agreed that the system captured all their attention, and 67% indicated that they agreed or completely agreed that the system makes them lose themselves in what they are performing.

In aspects related to **playfulness**, the average score was up to 5.54. Ninety percent indicated that they agree or completely agree that the system makes them curious, while seventy-six percent agreed or completely agreed that the overall feeling is fun. Fourteen percent indicated that the system does not allow them to be creative, that they do not have a sense of mystery, and that it does not encourage their imagination.

**Social** aspects were under four points (3.83), which is the least value. Forty-eight percent indicated that they completely disagreed or disagreed with the sentence "Makes me feel like I am socially involved", and thirty-three percent disagreed with the sentence "Gives me the feeling that I'm not on my own".

The results were adjusted to the type of experiment carried out. The environment has generated positive sensations in the fulfillment of the tasks, the moderate challenge, the feeling of having a teacher, immersion, and fun. However, the competition and the social aspects have not scored highly; this may be due to the fact that these aspects require continuous use of the system, which is not valued in this work, so although the experience includes these options, its results cannot be measured properly in this type of study.

Related to hypothesis 1, the users indicated that they feel motivated and feel curiosity about using the environment, they asked for more practices, and they expressed interest in including the system in a greater number of practical exercises. Then, hypothesis 1 was confirmed by the obtained results in this study design.

## 5.2. Usability

The second hypothesis of the work is related to the usability of the system, for which the SUS questionnaire was used. A SUS score of 68 or greater would be considered above average; in the data shown in Table 1, it can be seen that the virtual environment was evaluated with an 80. More than 95% (21/22) scored greater than this average. Eighty-five percent (18/22) indicated that the system was easy to use (with a score of 4 or 5), and ninety-five percent (21/22) indicated that various functions in this system were well integrated. No one indicated with a score higher than two that the system was unnecessarily complex (1/22) or inconsistent (0/22). Sixty-six percent (14/22) have felt very confident using the system.

**Table 1.** System usability scale results.

| Question | Mean | SD |
|---|---|---|
| I think that I would like to use this system frequently. | 3.48 | 1.17 |
| I found the system unnecessarily complex. | 1.29 | 0.56 |
| I thought the system was easy to use. | 4.24 | 1.00 |
| I think that I would need the support of a technical person to be able to use this system. | 1.86 | 0.85 |
| I found the various functions in this system were well integrated. | 4.43 | 0.60 |
| I thought there was too much inconsistency in this system. | 1.24 | 0.44 |
| I would imagine that most people would learn to use this system very quickly. | 3.95 | 1.07 |
| I found the system very cumbersome to use. | 2.05 | 1.40 |
| I felt very confident using the system. | 4.10 | 0.89 |
| I needed to learn a lot of things before I could get going with this system. | 1..38 | 0.92 |

Regarding Hypothesis 2, the results show that the system seems accessible enough and generates enough confidence for users to integrate it into their practical training routines. Therefore, this hypothesis could be considered confirmed by the results of this study design.

## 5.3. Sickness Feelings

The third working hypothesis is that the virtual environment does not produce negative effects on users. For this, the SSQ was used; Figure 12 shows the obtained results.

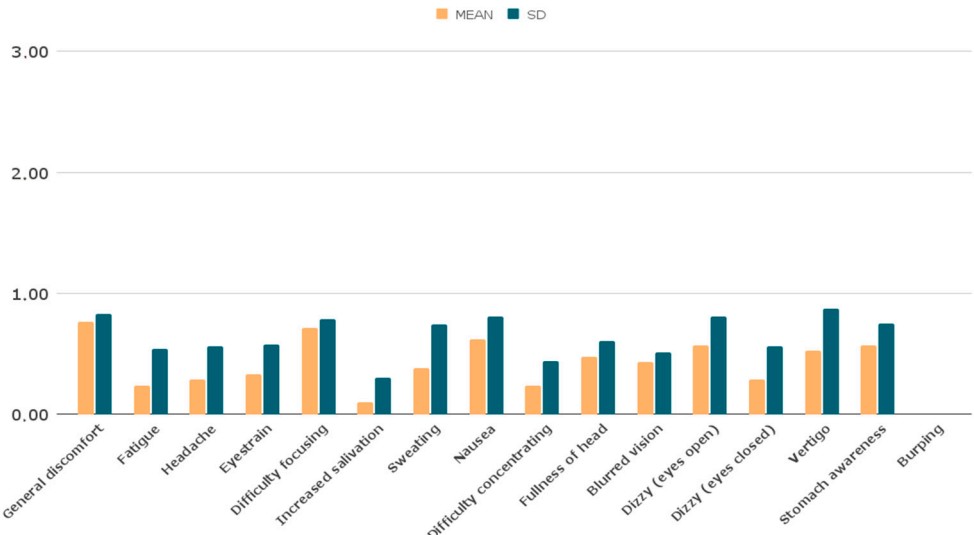

**Figure 12.** Simulator sickness questionnaire results.

The mean data for the SSQ, which can be "none" (0), "slight" (1), "moderate" (2) or "severe" (3), never exceeded "sight". General discomfort received a value higher (0.76), with just one participant indicating "severe", followed by difficulty focusing, where 18% indicated moderate or severe. "Fatigue", "Headache", "Increased salivation", "Difficulty concentrating", "Dizzy (eyes closed)", and "Burping" received average values under 0.3.

These data confirm hypothesis three that the virtual environment can be used with a low risk of discomfort during the period necessary to carry out practical training.

Note that the confirmation of the hypothesis is based on the results with a small sample. Therefore, further studies are needed to confirm the hypothesis generally.

## 6. Conclusions

This article presents a playful experience for the practice of aircraft maintenance, designed with the aim of facilitating training and coaching in this field of learning. As hypotheses, it has been supposed that the designed gamification processes encourage the practice of the designed task, that the usability of this playful experience facilitates the learning of the task, and, finally, that its use does not produce dizziness in users. Users have to overcome a series of tasks to bring theoretical concepts closer to practice. In order to validate each of these hypotheses, a test was designed that ends with the user answering a questionnaire: one for each hypothesis raised. The questions included in the questionnaires are based on scientific articles that provide the validity of the results.

The literature that evaluates the use of virtual or augmented reality for aircraft maintenance or the manipulation of other types of machinery continues to increase in quantity. Many of them focus on evaluating efficiency from a pedagogical point of view [17], while others go beyond efficiency and evaluate aspects such as presence [13]. In this work, we sought to analyze other components such as usability, symptoms of discomfort, or gameplay. Although this type of evaluation has not been found in aircraft maintenance, the results obtained are similar to those of works that apply it to other industrial or medical fields.

The analysis of the data endorses the good results of the experience, corroborating the hypotheses originally established. According to the results obtained from the GAME-FULQUEST, the gamification of the gaming experience encourages the practice of the processes required in aircraft maintenance. Users also confirmed that the usability of the designed play experience facilitates the practice of aircraft maintenance tasks, giving satisfactory answers to the SUS questionnaire that was offered after managing the proposed VR experience. Finally, the third hypothesis, which states that the VR experience does not produce any discomfort for users during the experience, has been endorsed by the results obtained with the SSQ. Although there are promising results, it is important to note that

the evaluation was conducted with a small sample size. As a result, it is crucial to stress the necessity of a more rigorous evaluation of the experience.

However, the experience currently includes very few tasks associated with aircraft maintenance work. We are currently working on improving the shortcomings detected in the application by users and on adding more operations that are considered essential for maintenance work. We are also studying the possibility of adding artificial intelligence to the experience in order to be able to add a chatbot to manage the doubts and answers of the students in the case of e-learning.

**Author Contributions:** Conceptualization, I.R. and I.M.; methodology, Á.G.-C.; software, A.T.; validation, Á.G.-C., A.T., I.M. and I.R.; formal analysis, Á.G.-C., I.M. and I.R.; investigation, Á.G.-C., I.M. and I.R.; resources, A.T.; data curation, Á.G.-C., A.T., I.M. and I.R.; writing—original draft preparation, I.M. and I.R.; writing—review and editing, I.M. and I.R.; visualization, A.T.; supervision, I.M. and I.R.; project administration, Á.G.-C.; funding acquisition, I.R. All authors have read and agreed to the published version of the manuscript.

**Funding:** Research was supported by the e-DIPLOMA project (project number 101061424), funded by the European Union. Views and opinions expressed are, however, those of the authors only and do not necessarily reflect those of the European Union or the European European Research Executive Agency (REA). Neither the European Union nor the granting authority can be held responsible for them.

**Institutional Review Board Statement:** The study was conducted in accordance with the Declaration of Helsinki and approved by the Ethics Committee of the Universitat Jaume I (code CEISH/28/2022).

**Informed Consent Statement:** Informed consent was obtained from all subjects involved in the study. Written informed consent has been obtained from the participant(s) to publish this paper.

**Data Availability Statement:** All the results extracted from the study are publicly available in CSV format in the 344 repository https://github.com/initUJI/airport-paper-data-set (accessed on 12 April 2023).

**Conflicts of Interest:** The authors declare no conflict of interest.

## Appendix A. GAMEFULQUEST

Please indicate how much you agree with the following statements, regarding your feelings while using the VR playful experience related to the learning of aircraft maintenance. Overall, the VR playful experience related to the learning of aircraft maintenance.

**Table A1.** GAMEFULQUEST Accomplishment section.

| | Strongly Disagree | Disagree | Somewhat Disagree | Neither Agree Nor Disagree | Somewhat Agree | Agree | Strongly Agree |
|---|---|---|---|---|---|---|---|
| Makes me feel that I need to complete things | | | | | | | |
| Pushes me to strive for accomplishments | | | | | | | |
| Inspires me to maintain my standards of performance | | | | | | | |
| Makes me feel that success comes through accomplishments | | | | | | | |
| Makes me strive to take myself to the next level | | | | | | | |
| Motivates me to progress and get better | | | | | | | |

**Table A1.** *Cont.*

| | Strongly Disagree | Disagree | Somewhat Disagree | Neither Agree Nor Disagree | Somewhat Agree | Agree | Strongly Agree |
|---|---|---|---|---|---|---|---|
| Makes me feel like I have clear goals | | | | | | | |
| Gives me the feeling that I need to reach goals | | | | | | | |

**Table A2.** GAMFEULQUEST Challenge section.

| | Strongly Disagree | Disagree | Somewhat Disagree | Neither Agree Nor Disagree | Somewhat Agree | Agree | Strongly Agree |
|---|---|---|---|---|---|---|---|
| Makes me push my limits | | | | | | | |
| Drives me in a good way to the brink of wanting to give up | | | | | | | |
| Pressures me in a positive way by its high demands | | | | | | | |
| Challenges me | | | | | | | |
| Calls for a lot of effort in order for me to be successful | | | | | | | |
| Motivates me to do things that feel highly demanding | | | | | | | |
| Makes me feel like I continuously need to improve in order to do well | | | | | | | |
| Makes me work at a level close to what I am capable of | | | | | | | |

**Table A3.** GAMFEULQUEST Competition section.

| | Strongly Disagree | Disagree | Somewhat Disagree | Neither Agree Nor Disagree | Somewhat Agree | Agree | Strongly Agree |
|---|---|---|---|---|---|---|---|
| Feels like participating in a competition | | | | | | | |
| Inspires me to compete | | | | | | | |
| Involves me by its competitive aspects | | | | | | | |
| Makes me want to be in first place | | | | | | | |
| Makes victory feel important | | | | | | | |
| Feels like being in a race | | | | | | | |
| Makes me feel that I need to win to succeed | | | | | | | |

**Table A4.** GAMFEULQUEST Guided section.

| | Strongly Disagree | Disagree | Somewhat Disagree | Neither Agree Nor Disagree | Somewhat Agree | Agree | Strongly Agree |
|---|---|---|---|---|---|---|---|
| Makes me feel guided | | | | | | | |
| Gives me a sense of being directed | | | | | | | |
| Makes me feel like someone is keeping me on track | | | | | | | |
| Gives me the feeling that I have an instructor | | | | | | | |

**Table A4.** *Cont.*

| | Strongly Disagree | Disagree | Somewhat Disagree | Neither Agree Nor Disagree | Somewhat Agree | Agree | Strongly Agree |
|---|---|---|---|---|---|---|---|
| Gives me the sense I am getting help to be structured | | | | | | | |
| Gives me a sense of knowing what I need to do to do better | | | | | | | |
| Gives me useful feedback so I can adapt | | | | | | | |

**Table A5.** GAMFEULQUEST Immersion section.

| | Strongly Disagree | Disagree | Somewhat Disagree | Neither Agree Nor Disagree | Somewhat Agree | Agree | Strongly Agree |
|---|---|---|---|---|---|---|---|
| Gives me the feeling that time passes quickly | | | | | | | |
| Grabs all of my attention | | | | | | | |
| Gives me a sense of being separated from the real world | | | | | | | |
| Makes me lose myself in what I am doing | | | | | | | |
| Makes my actions seem to come automatically | | | | | | | |
| Causes me to stop noticing when I get tired | | | | | | | |
| Causes me to forget about my everyday concerns | | | | | | | |
| Makes me ignore everything around me | | | | | | | |
| Gets me fully emotionally involved | | | | | | | |

**Table A6.** GAMFEULQUEST Playfulness section.

| | Strongly Disagree | Disagree | Somewhat Disagree | Neither Agree Nor Disagree | Somewhat Agree | Agree | Strongly Agree |
|---|---|---|---|---|---|---|---|
| Gives me an overall playful experience | | | | | | | |
| Leaves room for me to be spontaneous | | | | | | | |
| Taps into my imagination | | | | | | | |
| Makes me feel that I can be creative | | | | | | | |
| Gives me the feeling that I explore things | | | | | | | |
| Gives me a feeling that I want to know what comes next | | | | | | | |
| Makes me feel like I discover new things | | | | | | | |
| Makes me ignore everything around me | | | | | | | |
| Appeals to my curiosity | | | | | | | |

**Table A7.** GAMFEULQUEST Social experience section.

| | Strongly Disagree | Disagree | Somewhat Disagree | Neither Agree Nor Disagree | Somewhat Agree | Agree | Strongly Agree |
|---|---|---|---|---|---|---|---|
| Gives me an overall playful experience | | | | | | | |
| Gives me the feeling that I'm not on my own | | | | | | | |
| Gives me a sense of social support | | | | | | | |
| Taps into my imagination | | | | | | | |
| Makes me feel like I am socially involved | | | | | | | |
| Gives me a feeling of being connected to others | | | | | | | |
| Feels like a social experience | | | | | | | |
| Gives me a sense of having someone to share my endeavors with | | | | | | | |
| Influences me through its social aspects | | | | | | | |
| Gives me a sense of being noticed for what I have achieved | | | | | | | |

## Appendix B. Simulator Sickness Questionnaire

Instructions: Indicate how much each symptom below is affecting you right now.

**Table A8.** Simulator Sickness Questionnaire.

| | | | | |
|---|---|---|---|---|
| General discomfort | None | Slight | Moderate | Severe |
| Fatigue | None | Slight | Moderate | Severe |
| Headache | None | Slight | Moderate | Severe |
| Eye strain | None | Slight | Moderate | Severe |
| Difficulty focusing | None | Slight | Moderate | Severe |
| Salivation increasing | None | Slight | Moderate | Severe |
| Sweating | None | Slight | Moderate | Severe |
| Nausea | None | Slight | Moderate | Severe |
| Difficulty concentrating | None | Slight | Moderate | Severe |
| « Fullness of the Head » | None | Slight | Moderate | Severe |
| Blurred vision | None | Slight | Moderate | Severe |
| Dizziness with eyes open | None | Slight | Moderate | Severe |
| Dizziness with eyes closed | None | Slight | Moderate | Severe |
| Vertigo | None | Slight | Moderate | Severe |
| Stomach awareness | None | Slight | Moderate | Severe |
| Burping | None | Slight | Moderate | Severe |

## Appendix C. System Usability Scale

**Table A9.** System Usability Scale.

| | | Strongly Disagree | | | | Strongly Agree |
|---|---|---|---|---|---|---|
| 1. | I think that I would like to use this system frequently. | 1 | 2 | 3 | 4 | 5 |
| 2. | I found the system unnecessarily complex. | 1 | 2 | 3 | 4 | 5 |
| 3. | I thought the system was easy to use. | 1 | 2 | 3 | 4 | 5 |

**Table A9.** *Cont.*

|  |  | Strongly Disagree |  |  |  | Strongly Agree |
|---|---|---|---|---|---|---|
| 4. | I think that I would need the support of a technical person to be able to use this system. | 1 | 2 | 3 | 4 | 5 |
| 5. | I found the various functions in this system were well integrated. | 1 | 2 | 3 | 4 | 5 |
| 6. | I thought there was too much inconsistency in this system. | 1 | 2 | 3 | 4 | 5 |
| 7. | I would imagine that most people would learn to use this system very quickly. | 1 | 2 | 3 | 4 | 5 |
| 8. | I found the system very cumbersome to use. | 1 | 2 | 3 | 4 | 5 |
| 9. | I felt very confident using the system. | 1 | 2 | 3 | 4 | 5 |
| 10. | I needed to learn a lot of things before I could get going with this system. | 1 | 2 | 3 | 4 | 5 |

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
