# Peer review of "Immersive Virtual-Reality System for Aircraft Maintenance Education: A Case Study"

_applsci, doi:10.3390/app13085043_

Round 1

Reviewer 1 Report

The authors address an extremely important topic. It is crucial to utilize all available technological tools to shorten training time, thereby increasing teaching efficiency. To make the work more comprehensive, it would be necessary to supplement the introduction with a comparison of available commercial and non-commercial solutions for the Immersive Virtual-Reality System for Aircraft Maintenance Education, demonstrating how the proposed solution differs from or is similar to them.

The title and abstract suggest a description of the entire process of creating an educational environment. To achieve this, information and analyses regarding user requirements, technical training requirements, and business analyses are necessary. Based on this, correlations can be made to determine which requirements have been fulfilled and which have not, and reasons can be given. Such a process of implementing an innovative training tool adds value. Otherwise, we can only evaluate the environment, and the conclusions are known from the literature on VR.

It should be explicitly stated that a lot of work has gone into designing the test environment and gamification process. However, the research itself raises some doubts. Firstly, the research sample is very small, so this fact should be clearly emphasized in the manuscript. Global conclusions cannot be drawn based on such a small research group.

In lines 48-55, the authors describe the goals of the work. Not all of them are documented in the manuscript. Each educational tool should be evaluated for effectiveness compared to the classic approach. The authors state that the goal is to offer a cost-effective and effective way of training in the technical service of aircraft. However, there are no business or effectiveness analyses in the work.

The content of the conducted surveys should be included in the attachments, and the selection and characteristics of the selected individuals for the survey should be described more fully.

Additionally, information about the source is missing in Figure 7.

Author Response

Dear Sir/Madam,

Thank you for your thorough review. We have incorporated your suggestions and submitted the revised version of our manuscript titled “Immersive Virtual-Reality System for Aircraft Maintenance Education: a case study”, to Applied Science for publication consideration.

We will respond to your feedback in the attached document, analyzing the modifications that we have made following your advice.

Reviewer 2 Report

In the Introduction section, please describe the specifics of training aircraft maintenance specialists: what are the specific goals and possible problems of this process, and what methods are traditionally deployed in here, and what are their specific strengths and weaknesses. In addition, please compare your approach with the traditional methods, while, again, pointing out its weaknesses and strengths. 

Please describe shortly why you chose Unity 3D video game engine and compare it with alternatives (very shortly).

On rows 305-306, please describe if the excluded persons were counted towards the "involved 22 participants" or not (i.e., please confirm both the initial number of potential participants and the final number of participants). 

Also, please describe in 4.1. if the participants were familiar with aircraft maintenance and what was their knowledge level (i.e., were they students of specific fields)? Please describe the background of their testing - was your experiment a part of some of their study courses (describe the course in one sentence - name, volume) or was it conducted as an extracurriculum activity? Also, were there any benefits for the participants (was the participation mandatory or voluntary, did it influence their grades)? Where did the experiment take place? Where were the questionnaires filled in - at school, at home, etc.? How long did it take to fill in the three surveys?

In 4.2 please describe the surveys in more details. 

In 4.2 describe how were the data evaluated - who did the analysis (your researchers - one researcher or several - and if several then how were the discrepancies solved)? Did you use a specific software for analysis, which software?

In 4.2 and 5 - when you describe your Likert scales, list the minimum and maximum values (the values that you use in processing the data). Please describe why you chose (why it was possible for you) to treat your data as "interval data" instead of "ordinal data".

In Conclusion, compare your results with other similar works, even if from other fields (not from aircraft maintenance). 

Author Response

(The authors gave the same response as above.)

Round 2

Reviewer 1 Report

Thank you for including my comments in the manuscript. 

Author Response

Thank you for your review and comments.